# Markovian Score Climbing: Variational Inference with KL(p||q)

**Christian A. Naesseth**
Columbia University, USA
christian.a.naesseth@columbia.edu

**Fredrik Lindsten**
Linköping University, Sweden
fredrik.lindsten@liu.se

**David Blei**
Columbia University, USA
david.blei@columbia.edu

## Abstract

Modern variational inference (VI) uses stochastic gradients to avoid intractable expectations, enabling large-scale probabilistic inference in complex models. VI posits a family of approximating distributions $q$ and then finds the member of that family that is closest to the exact posterior $p$. Traditionally, VI algorithms minimize the "exclusive Kullback-Leibler (KL)" $\mathrm{KL}(q \parallel p)$, often for computational convenience. Recent research, however, has also focused on the "inclusive KL" $\mathrm{KL}(p \parallel q)$, which has good statistical properties that makes it more appropriate for certain inference problems. This paper develops a simple algorithm for reliably minimizing the inclusive KL using stochastic gradients with vanishing bias. This method, which we call Markovian score climbing (MSC), converges to a local optimum of the inclusive KL. It does not suffer from the systematic errors inherent in existing methods, such as Reweighted Wake-Sleep and Neural Adaptive Sequential Monte Carlo, which lead to bias in their final estimates. We illustrate convergence on a toy model and demonstrate the utility of MSC on Bayesian probit regression for classification as well as a stochastic volatility model for financial data.

## 1 Introduction

Variational inference (VI) is an optimization-based approach for approximate posterior inference. It posits a family of approximating distributions $q$ and then finds the member of that family that is closest to the exact posterior $p$. Traditionally, VI algorithms minimize the "exclusive Kullback-Leibler (KL)" $\mathrm{KL}(q \parallel p)$ [28, 6], which leads to a computationally convenient optimization. For a restricted class of models, it leads to coordinate-ascent algorithms [20]. For a wider class, it leads to efficient computation of unbiased gradients for stochastic optimization [51, 58, 52]. However, optimizing the exclusive KL results in an approximation that underestimates the posterior uncertainty [42]. To address this limitation, VI researchers have considered alternative divergences [35, 15]. One candidate is the "inclusive KL" $\mathrm{KL}(p \parallel q)$ [22, 7, 19]. This divergence more accurately captures posterior uncertainty, but results in a challenging optimization problem.

In this paper, we develop Markovian score climbing (MSC), a simple algorithm for reliably minimizing the inclusive KL. Consider a valid Markov chain Monte Carlo (MCMC) method [55], a Markov chain whose stationary distribution is $p$. MSC iteratively samples the Markov chain $\mathbf{z}[k]$, and then uses those samples to follow the score function of the variational approximation $\nabla \log q(\mathbf{z}[k])$ with a Robbins-Monro step-size schedule [54]. Importantly, we allow the MCMC method to depend on the current variational approximation. This enables a gradual improvement

of the MCMC as the VI converges. We illustrate this link between the methods by using conditional importance sampling (CIS) or conditional sequential Monte Carlo (CSMC) [2].

Other VI methods have targeted the same objective, including reweighted wake-sleep (RWS) [7] and neural adaptive sequential Monte Carlo (SMC) [22]. However, these methods involve biased gradients of the inclusive KL, which leads to bias in their final estimates. In contrast, MSC provides consistent gradients for essentially no added cost while providing better variational approximations. MSC provably converges to an optimum of the inclusive KL.

In empirical studies, we demonstrate the convergence properties and advantages of MSC. First, we illustrate the systematic errors of the biased methods and how MSC differs on a toy skew-normal model. Then we compare MSC with expectation propagation (EP) and importance sampling (IS)-based optimization [7, 19] on a Bayesian probit classification example with benchmark data. Finally, we apply MSC and SMC-based optimization [22] to fit a stochastic volatility model on exchange rate data.

**Contributions.** The contributions of this paper are (i) developing Markovian score climbing, a simple algorithm that provably minimizes $\mathrm{KL}(p \parallel q)$; (ii) studying systematic errors in existing methods that lead to bias in their variational approximation; and (iii) empirical studies that confirm convergence and illustrates the utility of MSC.

**Related Work.** Much recent effort in VI has focused on optimizing cost functions that are not the exclusive KL divergence. For example Rényi divergences and $\chi$ divergence are studied in [35, 15]. The most similar to our work are the methods in [7, 22, 19], using IS or SMC to optimize the inclusive KL divergence. The RWS algorithm [7] uses IS both to optimize model parameters and the variational approximation. Neural adaptive SMC [22] jointly learn an approximation to the posterior and optimize the marginal likelihood of time series with gradients estimated by SMC. In [19] connections between importance weighted autoencoders [9], adaptive IS and methods like the RWS are drawn. These three works all rely on IS or SMC to estimate expectations with respect to the posterior. This introduces a systematic bias in the gradients that leads to a solution which is not a local optimum to the inclusive KL divergence. In [50] inference networks are learnt for data simulated from the model rather than observed data.

Another line of work studies the combination of VI with Monte Carlo (MC) methods. Salimans et al. [59] take inspiration from the MCMC literature to define their variational approximation. The method in [26] uses the variational approximation to improve Hamiltonian MC. Variational SMC [46, 34, 40] uses the SMC sample process itself to define an approximation to the posterior. Follow up work [33, 44] improve on variational SMC in various ways by using *twisting* [23, 25, 39]. Another approach takes a MC estimator of the marginal likelihood and turn it into a posterior approximation [16]. The method in [24] uses auxiliary variables to define a more flexible approximation to the posterior, then subsequently at test time apply MCMC. These methods all optimize a variational approximation based on MC methods to minimize the exclusive KL divergence. On the contrary, the method proposed in this paper minimizes the inclusive KL divergence. The method in [27] optimizes an initial approximation to the posterior in exclusive KL, then refines this with a few iterations of MCMC to estimate gradients with respect to the model parameters. Defining the variational approximation as an initial distribution to which a few steps of MCMC is applied, and then optimize a new contrastive divergence is done in [56]. This divergence is different from the inclusive KL and MCMC is used as a part of the variational approximation rather than gradient estimation. Another line of work studies combinations of MC and VI using amortization [36, 61, 62].

Using MC together with stochastic optimization, for e.g. maximum likelihood estimation of latent variable models, is studied in [21, 31, 1, 14]. In contrast the proposed method uses it for VI. Viewing MSC as a way to learn a proposal distribution for IS means it is related to the class of adaptive IS algorithms [8, 10, 17, 11]. We compare to IS/SMC-based optimization, as outlined in the background, in the experimental studies which can be considered to be special cases of adaptive IS/SMC.

Concurrent and independent work using MCMC to optimize the inclusive KL was studied in [48]. The difference with our work lies in the Markov kernels used, our focus on continuous latent variables, and our study of the impact of large-scale exchangeable data.

## 2   Background

Let $p(\mathbf{z}, \mathbf{x})$ be a probabilistic model for the latent (unobserved) variables $\mathbf{z}$ and data $\mathbf{x}$. In Bayesian inference the main concern is computing the posterior distribution $p(\mathbf{z}|\mathbf{x})$, the conditional distribution of the latent variables given the observed data. The posterior is $p(\mathbf{z}|\mathbf{x}) = p(\mathbf{z}, \mathbf{x})/p(\mathbf{x})$. The normalization constant is the marginal likelihood $p(\mathbf{x})$, computed by integrating (or summing) the joint model $p(\mathbf{z}, \mathbf{x})$ over all values of $\mathbf{z}$. For most models of interest, however, exactly computing the posterior is intractable, and we must resort to a numerical approximation.

### 2.1   Variational Inference with KL(p||q)

One approach to approximating the posterior is with VI. This turns the intractable problem of computing the posterior into an optimization problem that can be solved numerically. The idea is to first posit a variational family of approximating distributions $q(\mathbf{z}; \lambda)$, parametrized by $\lambda$. Then minimize a metric or divergence so that the variational approximation is close to the posterior, $q(\mathbf{z}; \lambda) \approx p(\mathbf{z}|\mathbf{x})$.

The most common VI objective is to minimize the exclusive KL, $\mathrm{KL}(q \,\|\, p)$. This objective is an expectation with respect to the approximating distribution $q$ that is convenient to optimize. But this convenience comes at a cost—the $q$ optimized to minimize $\mathrm{KL}(q \,\|\, p)$ will underestimate the variance of the posterior [15, 6, 60].

One way to mitigate this issue is to instead optimize the inclusive KL,

$$\mathrm{KL}(p(\mathbf{z}|\mathbf{x}) \,\|\, q(\mathbf{z}; \lambda)) := \mathbb{E}_{p(\mathbf{z}|\mathbf{x})}\left[\log p(\mathbf{z}|\mathbf{x}) - \log q(\mathbf{z}; \lambda)\right]. \tag{1}$$

This objective, though more difficult to work with, does not lead to underdispersed approximations. For too simplistic $q$ it might lead to approximations that stretch to cover $p$ putting mass even where $p$ is small, thus leading to poor predictive distributions. However, in the context of VI inclusive KL has motivated among others neural adaptive SMC [22], RWS [7], and EP [43]. This paper develops MSC, a new algorithm to minimize the inclusive KL divergence.

Minimizing eq. (1) is equivalent to minimizing the cross entropy $L_{\mathrm{KL}}(\lambda)$,

$$\min_{\lambda} L_{\mathrm{KL}}(\lambda) := \min_{\lambda} \mathbb{E}_{p(\mathbf{z}|\mathbf{x})}\left[-\log q(\mathbf{z}; \lambda)\right]. \tag{2}$$

The gradient w.r.t. the variational parameters is

$$g_{\mathrm{KL}}(\lambda) := \nabla L_{\mathrm{KL}}(\lambda) = \mathbb{E}_{p(\mathbf{z}|\mathbf{x})}\left[-s(\mathbf{z}; \lambda)\right], \tag{3}$$

where we define $s(\mathbf{z}; \lambda)$ to be the *score function*,

$$s(\mathbf{z}; \lambda) := \nabla_{\lambda} \log q(\mathbf{z}; \lambda). \tag{4}$$

Because the cross entropy is an expectation with respect to the (intractable) posterior, computing its gradient pointwise is intractable. Recent algorithms for solving eq. (2) focus on *stochastic gradient descent* [7, 22, 19].

### 2.2   Stochastic Gradient Descent with IS

We use stochastic gradient descent (SGD) in VI when the gradients of the objective are intractable. The SGD updates

$$\lambda_k = \lambda_{k-1} - \varepsilon_k \, \widehat{g}_{\mathrm{KL}}(\lambda_{k-1}), \tag{5}$$

converges to a local optimum of eq. (2) if the gradient estimate $\widehat{g}_{\mathrm{KL}}$ is unbiased, $\mathbb{E}\left[\widehat{g}_{\mathrm{KL}}(\lambda)\right] = g_{\mathrm{KL}}(\lambda)$, and the step sizes satisfy $\sum_k \varepsilon_k^2 < \infty$, $\sum_k \varepsilon_k = \infty$ [54, 32].

When the objective is the exclusive $\mathrm{KL}(q \,\|\, p)$, we can use score-function gradient estimators [51, 58, 52], reparameterization gradient estimators [53, 30], or combinations of the two [57, 45]. These methods provide unbiased stochastic gradients that can help find a local optimum of the exclusive KL.

However, we consider minimizing the inclusive $\mathrm{KL}(p \,\|\, q)$ eq. (1), for which gradient estimation is difficult. It requires an expectation with respect to the posterior $p$. One strategy is to use IS [55] to rewrite the gradient as an expectation with respect to $q$. Specifically, the gradient of the inclusive KL is proportional to

$$\nabla_{\lambda} L_{\mathrm{KL}}(\lambda) \propto -\mathbb{E}_{q(\mathbf{z}; \lambda)}\left[\frac{p(\mathbf{z}, \mathbf{x})}{q(\mathbf{z}; \lambda)} s(\mathbf{z}; \lambda)\right], \tag{6}$$

where the constant of proportionality $1/p(\mathbf{x})$ is independent of the variational parameters and will not affect the solution of the corresponding fixed point equation. This gradient is unbiased, but estimating it using standard MC methods can lead to high variance and poor convergence.

Another option [22, 7] is the self-normalized IS (or corresponding SMC) estimate

$$\nabla_\lambda L_{\mathrm{KL}}(\lambda) \approx -\sum_{s=1}^{S} \frac{w_s}{\sum_{r=1}^{S} w_r} s(\mathbf{z}^s ; \lambda), \tag{7}$$

where $w_s = p(\mathbf{z}^s, \mathbf{x})/q(\mathbf{z}^s ; \lambda)$, $\mathbf{z}^s \sim q(\mathbf{z}^s ; \lambda)$, and $s(\mathbf{z} ; \lambda) = \nabla_\lambda \log q(\mathbf{z} ; \lambda)$. However, eq. (7) is not unbiased. The estimator suffers from systematic error and, consequently, the fitted variational parameters are no longer optimal with respect to the original minimization problem in eq. (2). (See [47, 49, 55] for details about IS and SMC methods) MSC addresses this shortcoming, introducing an algorithm that provably converges to a solution of eq. (2). In the remainder of the paper IS refers to self-normalized IS.

## 3 Markovian Score Climbing

The key idea in MSC is to use MCMC methods to estimate the intractable gradient. Under suitable conditions on the algorithm, MSC is guaranteed to converge to a local optimum of $\mathrm{KL}(p \parallel q)$.

First, we discuss generic MCMC methods to estimate gradients in a SGD algorithm. Importantly, the MCMC method can depend on the current VI approximation which provides a tight link between MCMC and VI. Next we exemplify this connection by introducing CIS, an example Markov kernel that is a simple modification of IS, where the VI approximation is used as a proposal. The extra computational cost is negligible compared to the biased approaches discussed in section 2.2, CIS only generates a single extra categorical random variable per iteration. The corresponding extension to SMC, i.e. the CSMC kernel, is discussed in the supplement. Next, we discuss learning model parameters. Then, we show that the resulting MSC algorithm is exact in the sense that it converges asymptotically to a local optima of the inclusive KL divergence. Finally, we discuss large-scale data.

### 3.1 Stochastic Gradient Descent using MCMC

When using gradient descent to optimize the inclusive KL we must compute an expectation of the score function $s(\mathbf{z} ; \lambda)$ eq. (4) with respect to the true posterior. To avoid this intractable expectation we propose to use stochastic gradients estimated using samples generated from a MCMC algorithm, with the posterior as its stationary distribution. The key step to ensure convergence, without having to run an infinite inner loop of MCMC updates, is to *not* re-initialize the Markov chain at each step $k$. Instead, the sample $\mathbf{z}[k-1]$ used to estimate the gradient at step $k-1$ is passed to a Markov kernel $\mathbf{z}[k] \sim M(\cdot \mid \mathbf{z}[k-1])$, with the posterior as its stationary distribution, to get an updated $\mathbf{z}[k]$ that is then used to estimate the current gradient, i.e. the score $s(\mathbf{z}[k] ; \lambda)$. This leads to a Markovian stochastic approximation algorithm [21], where the noise in the gradient estimate is Markovian. Because we are moving in an ascent direction of the score function at each iteration and using MCMC, we refer to the method developed in this paper as Markovian score climbing.

It is not a requirement that the Markov kernel $M$ is independent of the variational parameters $\lambda$. In fact it is key for best performance of MSC that we use the variational approximation to define the Markov chain. We summarize MSC in algorithm 1.

Next, we discuss CIS [2, 47], an example Markov kernel with adaptation that is a simple modifications of its namesake IS. The corresponding extension to SMC, the CSMC kernel [2, 47], is discussed in the supplement. Using these Markov kernels to estimate gradients, rather than IS and SMC [22, 7], lead to algorithms that are simple modifications of their non-conditional counterparts but provably converge to a local optimum of the inclusive KL divergence.

### 3.2 Conditional Importance Sampling

CIS is an IS-based Markov kernel with $p(\mathbf{z} \mid \mathbf{x})$ as its stationary distribution [2, 47]. It modifies the classical IS algorithm by retaining one of the samples from the previous iteration, the so-called *conditional sample*. Each iteration consists of three steps: generate new samples from a proposal,

---

**Algorithm 1:** Markovian Score Climbing

---

**Input** : Markov kernel $M(\mathbf{z}'\,|\,\mathbf{z}\,;\lambda)$ with stationary distribution $p(\mathbf{z}\,|\,\mathbf{x})$, variational family $q(\mathbf{z}\,;\lambda)$, initial $\lambda_0$, initial $\mathbf{z}[0]$, step size sequence $\varepsilon_k$, and number of iterations $K$.

**Output** : $\lambda_K \approx \lambda^\star$.

---

1 **for** $k = 1, \ldots, K$ **do**
2 $\quad$ Sample $\mathbf{z}[k] \sim M(\cdot\,|\,\mathbf{z}[k-1]\,;\lambda_{k-1})$
3 $\quad$ Compute $s(\mathbf{z}[k]\,;\lambda_{k-1}) = \nabla_\lambda \log q(\mathbf{z}[k]\,;\lambda_{k-1})$
4 $\quad$ Set $\lambda_k = \lambda_{k-1} + \varepsilon_k s(\mathbf{z}[k]\,;\lambda_{k-1})$
5 **end**

---

compute weights, and then update the conditional sample for the next iteration. We explain in detail below.

First, set the first proposed sample to be equal to the conditional sample from the previous iteration, i.e. $\mathbf{z}^1 = \mathbf{z}[k-1]$, and propose the remaining $S-1$ samples from a proposal distribution $\mathbf{z}^i \sim q(\mathbf{z}\,;\lambda)$, $\quad i = 2, \ldots, S$. The proposal does not necessarily need to be equal to the variational approximation, a common option is to use the model prior $p(\mathbf{z})$. However, we will in the remainder of this paper assume that the variational approximation $q(\mathbf{z}\,;\lambda)$ is used as the proposal. This provides a link between the MCMC proposal and the current VI approximation. Then, compute the importance weights for all $S$ samples, including the conditional sample. The importance weights for $i = 1, \ldots, S$ are $w^i = {p(\mathbf{z}^i,\mathbf{x})}/{q(\mathbf{z}^i;\lambda)}$, $\bar{w}^i = {w^i}/{\sum_{j=1}^S w^j}$. Finally, generate an updated conditional sample by picking one of the proposed values with probability proportional to its (normalized) weight, i.e., $\mathbf{z}[k] = \mathbf{z}^J$, where $J$ is a discrete random variable with probability $\mathbb{P}(J = j) = \bar{w}^j$.

Iteratively repeating this procedure constructs a Markov chain with the posterior $p(\mathbf{z}\,|\,\mathbf{x})$ as its stationary distribution [2, 47]. With this it is possible to attain an estimate of the (negative) gradient w.r.t. the variational parameters of eq. (2):

$$s(\mathbf{z}[k]\,;\lambda) = \nabla_\lambda \log q(\mathbf{z}[k]\,;\lambda), \tag{8}$$

where $\mathbf{z}[k]$ is the conditional sample retained at each iteration of the CIS algorithm. Another option is to make use of all samples at each iteration, i.e. the Rao-Blackwellized estimate, $\widehat{g}_{\mathrm{KL}}(\lambda) = \sum_{i=1}^S \bar{w}^i s(\mathbf{z}^i\,;\lambda)$. We summarize one full iteration of the CIS in algorithm 2.

---

**Algorithm 2:** Conditional Importance Sampling

---

**Input** : Model $p(\mathbf{z},\mathbf{x})$, proposal $q(\mathbf{z}\,;\lambda)$, conditional sample $\mathbf{z}[k-1]$, and total number of internal samples $S$.

**Output** : $\mathbf{z}[k] \sim M(\cdot\,|\,\mathbf{z}[k-1]\,;\lambda)$, updated conditional sample.

---

1 Set $\mathbf{z}^1 = \mathbf{z}[k-1]$
2 Sample $\mathbf{z}^i \sim q(\mathbf{z}\,;\lambda)$ for $i = 2, \ldots, S$
3 Compute $w^i = {p(\mathbf{z}^i,\mathbf{x})}/{q(\mathbf{z}^i;\lambda)}$, $\bar{w}^i = {w^i}/{\sum_{j=1}^S w^j}$ for $i = 1, \ldots, S$
4 Sample $J$ with probability $\mathbb{P}(J = j) \propto \bar{w}^j$
5 Set $\mathbf{z}[k] = \mathbf{z}^J$

---

### 3.3 Model Parameters

If the probabilistic model has unknown parameters $\theta$ one solution is to assign them a prior distribution, include them in the latent variable $\mathbf{z}$, and apply the method outlined above to approximate the posterior. However, an alternative solution is to use the maximum likelihood (ML) principle and optimize the marginal likelihood, $p(\mathbf{x}\,;\theta)$, jointly with the approximate posterior, $q(\mathbf{z}\,;\lambda)$. We propose to use Markovian score climbing based on the Fisher identity of the gradient

$$g_{\mathrm{ML}}(\theta) = \nabla_\theta \log p(\mathbf{x}\,;\theta) = \nabla_\theta \log \int p(\mathbf{z},\mathbf{x}\,;\theta)\,\mathrm{d}\mathbf{z} = \mathbb{E}_{p_\theta(\mathbf{z}|\mathbf{x})}\left[\nabla_\theta \log p(\mathbf{z},\mathbf{x}\,;\theta)\right]. \tag{9}$$

With a Markov kernel $M(\mathbf{z}[k]\,|\,\mathbf{z}[k-1]\,;\,\theta,\lambda)$, with the posterior distribution $p(\mathbf{z}\,|\,\mathbf{x}\,;\,\theta)$ as its stationary distribution, the approximate gradient is $\widehat{g}_{\mathrm{ML}}(\theta) = \nabla_\theta \log p(\mathbf{z}[k],\mathbf{x}\,;\,\theta)$.

The MSC algorithm for maximization of the log-marginal likelihood, with respect to $\theta$, and minimization of the inclusive KL divergence, with respect to $\lambda$, is summarized in algorithm 3. Using MSC *only* for ML estimation of $\theta$, with a fixed Markov kenel $M$ and *without* the VI steps on lines 13 and 15, is equivalent to the MCMC ML method in [21].

---

**Algorithm 3:** Markovian Score Climbing with ML

**Input** : Markov kernel $M(\mathbf{z}'\,|\,\mathbf{z}\,;\,\theta,\lambda)$ with stationary distribution $p(\mathbf{z}\,|\,\mathbf{x}\,;\,\theta)$, variational family $q(\mathbf{z}\,;\,\lambda)$, initial $\lambda_0, \mathbf{z}[0], \theta_0$, step size sequences $\varepsilon_k, \epsilon_k$, and iterations $K$.
**Output** : $\lambda_K \approx \lambda^\star$, $\theta_K \approx \theta^\star$.

1 **for** $k = 1,\ldots,K$ **do**
2      Sample $\mathbf{z}[k] \sim M(\cdot\,|\,\mathbf{z}[k-1]\,;\,\theta_{k-1},\lambda_{k-1})$
3      Compute $s(\mathbf{z}[k]\,;\,\lambda_{k-1}) = \nabla_\lambda \log q(\mathbf{z}[k]\,;\,\lambda_{k-1})$
4      Compute $\widehat{g}_{\mathrm{ML}}(\theta_{k-1}) = \nabla_\theta \log p(\mathbf{z}[k],\mathbf{x}\,;\,\theta_{k-1})$
5      Set $\lambda_k = \lambda_{k-1} + \varepsilon_k s(\mathbf{z}[k]\,;\,\lambda_{k-1})$
6      Set $\theta_k = \theta_{k-1} + \epsilon_k \widehat{g}_{\mathrm{ML}}(\theta_{k-1})$
7 **end**

---

## 3.4 The Convergence of MSC

One of the main benefits of MSC is that it is possible, under certain regularity conditions, to ensure that the variational parameter estimate $\lambda_K$ as provided by algorithm 1 converges to a local optima of the inclusive KL divergence as the number of iterations $K$ tend to infinity. We formalize the convergence result in proposition 1. The result is an application of [21, Theorem 1] and based on [5, Theorem 3.17, page 304]. The proof is found in the Supplement.

**Proposition 1.** *Assume that C1–C6, detailed in the supplement, hold. If $\lambda_k$ for $k \geq 1$ defined by algorithm 1 is a bounded sequence and almost surely visits a compact subset of the domain of attraction of $\lambda^\star$ infinitely often, then*

$$\lambda_k \to \lambda^\star, \quad \text{almost surely.}$$

## 3.5 MSC on Large-Scale Data

If the dataset $\mathbf{x} = (x_1,,\ldots,x_n)$ is large it might be impractical to evaluate the full likelihood at each step and it would be preferable to consider only a subset of the data at each iteration. Variational inference based on the exclusive KL, $\mathrm{KL}(q\,\|\,p)$, is scalable in the sense that it works by subsampling datasets both for *exchangeable data*, $p(\mathbf{x}) = \mathbb{E}_{p(\mathbf{z})}\big[\prod_{i=1}^{n} p(x_i\,|\,\mathbf{z})\big]$, as well as for *independent and identically distributed data* (iid), $p(\mathbf{x}) = \prod_{i=1}^{n} p(x_i) = \prod_{i=1}^{n} \mathbb{E}_{p(z_i)}[p(x_i\,|\,z_i)]$ where $\mathbf{z} = (z_1,\ldots,z_n)$. For the exclusive KL divergence subsampling is straightforward; the likelihood enters as a sum of the individual log-likelihood terms for all datapoints whether the data is iid or exchangeable, and a simple unbiased estimate can be constructed by sampling one (or a few) datapoints to evaluate at each iteration. However, for the inclusive KL divergence the large-scale data implications for the two settings are less clear and we discuss each below.

Often in the literature [7, 9, 14, 48] applications assumes the data is generated *iid* and achieve scalability through use of subsampling and amortization. In fact, MSC can potentially scale just as well as other algorithms to large datasets when data is assumed iid $x_i \sim p(x)$, $i = 1,\ldots,n$. Instead of minimizing $\mathrm{KL}(p(z\,|\,x)\,\|\,q(z\,;\,\lambda))$ wrt $\lambda$ for each $x = x_i$, we consider minimizing $\mathrm{KL}\big(p(x)p(z|x)\,\|\,p(x)q(z\,|\,\lambda_\eta(x))\big)$ wrt $\eta$ where $\lambda_\eta(x)$ is an inference network (amortization). If $q(z\,|\,\lambda_\eta(x))$ is flexible enough the posterior $p(z\,|\,x)$ is the optimal solution to this minimization problem. Stochastic gradient descent can be performed by noting that

$$\nabla_\eta \mathrm{KL}\big(p(x)p(z|x)\,\|\,p(x)q(z\,|\,\lambda_\eta(x))\big) = 0 + \mathbb{E}_{p(x)p(z|x)}\big[-\nabla_\eta \log q(z|\lambda_\eta(x)\big]$$

$$\approx \frac{1}{n}\sum_{i=1}^{n} \mathbb{E}_{p(z|x_i)}\big[-\nabla_\eta \log q(z|\lambda_\eta(x_i))\big],$$

where the approximation is directly amenable to data subsampling. We leave the formal study of this approach for future work.

For *exchangeable data* the likelihood enters as a product and subsampling is difficult in general. Standard MCMC kernels require evaluation of the complete likelihood at each iteration, which means that the method proposed in this paper likewise must evaluate all the data points at each iteration of algorithm 1. An option is to follow [35, 15] using *subset average likelihoods*. In appendix A.1 we prove that this approach leads to systematic errors that are difficult to quantify. It does not minimize the inclusive KL from $p$ to $q$, rather it minimizes the KL divergence from a *perturbed* posterior $\tilde{p}$ to $q$. A potential remedy to this issue, that we leave for future work, is to consider approximate MCMC (with theoretical guarantees) reviewed in e.g. [4, 3].

## 4 Empirical Evaluation

We illustrate convergence on a toy model and demonstrate the utility of MSC on Bayesian probit regression for classification as well as a stochastic volatility model for financial data. The studies show that MSC (i) converges to the true solution whereas the biased methods do not; (ii) achieves similar predictive performance as EP and IS on regression while being more robust to the choice of sample size $S$; and (iii) learns superior or as good stochastic volatility models as SMC. Code is available at github.com/blei-lab/markovian-score-climbing.

### 4.1 Skew Normal Distribution

We illustrate the impact of the biased gradients discussed in section 2.2 on a toy example. Let $p(\mathbf{z}|\mathbf{x})$ be a scalar skew normal distribution with location, scale and shape parameters $(\xi, \omega, \alpha) = (0.5, 2, 5)$. We let the variational approximation be a family of normal distributions $q(\mathbf{z}; \lambda) = \mathcal{N}(\mathbf{z}; \mu, \sigma^2)$. For this choice of posterior and approximating family it is possible to compute the analytical solution for the inclusive KL divergence; it corresponds to matching the moments of the variational approximation and the posterior distribution. In fig. 1 we show the results of SGD when using the biased gradients from eq. (7), i.e. using self-normalized IS to estimate the gradients, and MSC (this paper) as described in section 3. We set the number of samples to $S = 2$. We can see how the biased gradient leads to systematic errors when estimating the variational parameters, whereas MSC obtains the true solution. Increasing the number of samples for the estimator in eq. (7) will lower the bias, and in the limit of infinite samples $S$ it is exact. However, for non-toy problems it is likely very difficult to know what is a sufficient number of samples to get an "acceptable bias" in the VI solution. MSC, on the other hand, provides consistent estimates of the variational parameters even with small number of samples. Note that

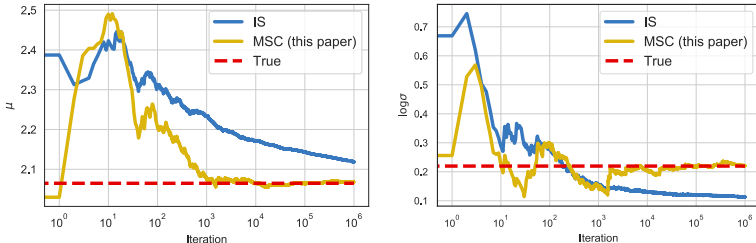

**Figure 1:** MSC converges to the true solution, while the biased IS approach does not. Example of learnt variational parameters for IS- and MSC-based gradients of the inclusive KL, as well as true parameters. Gaussian approximation to a skew normal distribution. Iterations in log-scale.

the biased IS-gradients results in an *underestimation* of the variance. One of the main motivations for using inclusive KL as optimization objective is to avoid such underestimation of uncertainty. This example shows that when the inclusive KL is optimized with *biased gradients* the solution can no longer be trusted in this respect. The gradients for Rényi- and $\chi$ divergences used in e.g. [35, 15] suffer from a similar bias. The supplement provides a $\chi$ divergence analogue to fig. 1.

### 4.2 Bayesian Probit Regression

Probit regression is commonly used for binary classification in machine learning and statistics. The Bayesian probit regression model assigns a Gaussian prior to the parameters. The prior and

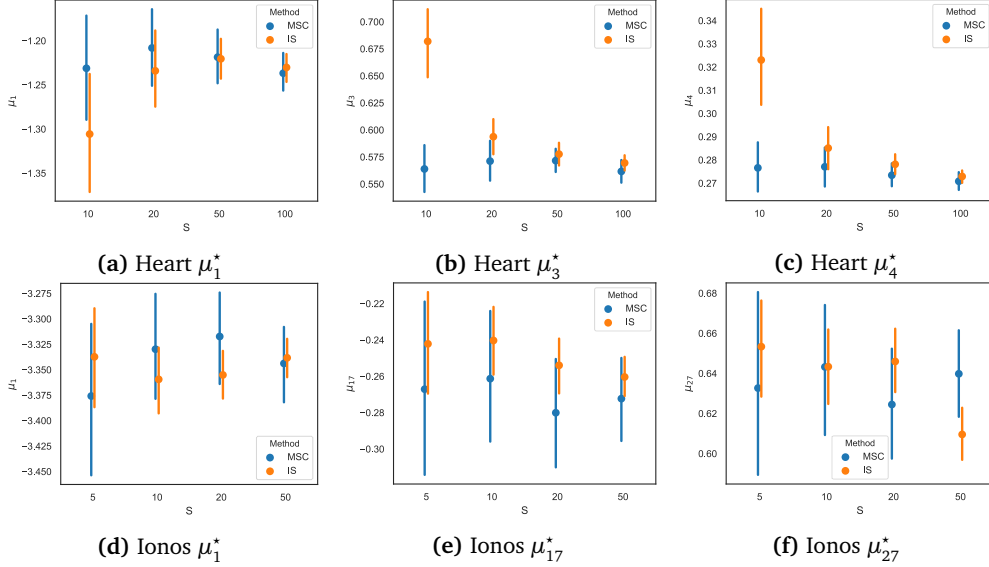

**Figure 2:** MSC is more robust to the number of samples $S$. The fitted mean parameter $\mu^\star$, for three representative dimensions of $\mathbf{z}$, of MSC (this paper) and IS (cf. [7]) on the Ionos and Heart datasets. The error bars corresponds to 100 random initializations.

likelihood are $p(\mathbf{z}) = \mathcal{N}(\mathbf{z}; 0, I)$, $\mathbb{P}(\mathbf{y}_t = y \mid \mathbf{z}, \mathbf{x}_t) = \Phi(\mathbf{x}_t^\top \mathbf{z})^y \left(1 - \Phi(\mathbf{x}_t^\top \mathbf{z})\right)^{1-y}$, where $y \in \{0, 1\}$ and $\Phi(\cdot)$ is the cumulative distribution function of the normal distribution. We apply the model for prediction in several UCI datasets [18]. We let the variational approximation be a Gaussian distribution $q(\mathbf{z}; \lambda) = \mathcal{N}(\mathbf{z}; \mu, \Sigma)$, where $\Sigma$ is a diagonal covariance matrix. We compare MSC (this paper) with the biased IS-based approach (cf. eq. (7) and [7]) and EP [43] that minimizes the inclusive KL locally. For SGD methods we use adaptive step-sizes [29].

Table 1 illustrates the predictive performance of the fitted model on held-out test data. The results where generated by splitting each dataset 100 times into 90% training and 10% test data, then computing average prediction error and its standard deviation. MSC performs as well as EP which is particularly well suited to this problem. However, EP requires more model-specific derivations and can be difficult to implement when the moment matching subproblem can not be solved in closed form. In these experiments the bias introduced by IS does not significantly impact the predictive performance compared to MSC.

| Dataset | EP [43] | IS [7] | MSC (adaptive) | MSC (prior) |
|---|---|---|---|---|
| Pima | $0.227 \pm 0.048$ | $0.229 \pm 0.047$ | $0.227 \pm 0.046$ | $0.456 \pm 0.093$ |
| Ionos | $0.115 \pm 0.053$ | $0.115 \pm 0.054$ | $0.117 \pm 0.053$ | $0.182 \pm 0.070$ |
| Heart | $0.161 \pm 0.066$ | $0.163 \pm 0.066$ | $0.160 \pm 0.063$ | $0.342 \pm 0.11$ |

**Table 1:** Test error for Bayesian probit regression; lower is better. Estimated using EP [43], IS (cf. [7]), and MSC (this paper) with proposal $q(\mathbf{z}; \lambda)$ (adaptive) or $p(\mathbf{z})$ (prior) for 3 UCI datasets. Predictive performance is comparable, but MSC is more robust and generically applicable.

We compare how the approximations based on MSC and IS are affected by the number of samples $S$ at each iteration. In fig. 2 we plot the mean value $\mu^\star$ based on 100 random initializations for several values of $S$ on the Heart and Ionos datasets. The MSC is more robust to the choice of $S$, converging to similar mean values for all the choices of $S$ in this example. For the Heart dataset, IS clearly struggles with a bias for low values of the number of samples $S$.

### 4.3 Stochastic Volatility

The stochastic volatility model is commonly used in financial econometrics [12]. The model is $p(\mathbf{z}_0; \theta) = \mathcal{N}\left(\mathbf{z}_0; 0, \frac{\sigma^2}{1-\phi^2}\right)$, $p(\mathbf{z}_t \mid \mathbf{z}_{t-1}; \theta) = \mathcal{N}(\mathbf{z}_t; \mu + \phi(\mathbf{z}_{t-1} - \mu), \sigma^2)$, $p(\mathbf{x}_t \mid \mathbf{z}_t; \theta) = \mathcal{N}(\mathbf{x}_t; 0, \beta \exp(\mathbf{z}_t))$, where the parameters are constrained as follows $\theta = \left(\sigma^2, \phi, \mu, \beta\right) \in \mathbb{R}_+ \times (-1, 1) \times \mathbb{R} \times \mathbb{R}_+$. Both the posterior distribution and log-marginal likelihood are intractable so we make use of algorithm 3 as outlined in section 3.3 with the CSMC kernel described in the supple-

ment. The proposal distributions are $q(\mathbf{z}_0 \,;\, \theta, \lambda_0) \propto p(\mathbf{z}_0 \,;\, \theta) e^{-\frac{1}{2}\Lambda_0 z_0^2 + \nu_0 \mathbf{z}_0}$, $q(\mathbf{z}_t \,|\, \mathbf{z}_{t-1} \,;\, \theta, \lambda_t) \propto p(\mathbf{z}_t \,|\, \mathbf{z}_{t-1} \,;\, \theta) e^{-\frac{1}{2}\Lambda_t z_t^2 + \nu_t \mathbf{z}_t}$, with variational parameters $\lambda_t = (\nu_t, \Lambda_t) \in \mathbb{R} \times \mathbb{R}_+$.

We compare MSC with the SMC-based approach [22] using adaptive step-size [29]. We study monthly returns over 10 years (9/2007 to 8/2017) for the exchange rate of 18 currencies with respect to the US dollar. The data is obtained from the Federal Reserve System. In fig. 3 we illustrate the difference between the log-marginal likelihood obtained by the two methods, $\log p(\mathbf{x} \,;\, \theta^\star_{\mathrm{MSC}}) - \log p(\mathbf{x} \,;\, \theta^\star_{\mathrm{SMC}})$. We learn the model and variational parameters using $S = 10$ particles for both methods, and estimate the log-marginal likelihood after convergence using $S = 10,000$. The log-marginal likelihood obtained by MSC is significantly better than SMC for several of the datasets.

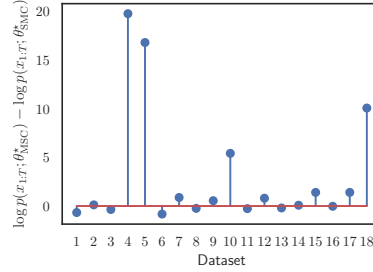

**Figure 3:** Difference in log-marginal likelihood values for parameters learnt by MSC (this paper) and SMC [22]. The likelihood obtained by MSC, on average, is superior to or as good as that obtained by SMC.

## 5    Conclusions

In VI, the properties of the approximation $q$, to the posterior $p$, depends on the choice of divergence that is minimized. The most common choice is the exclusive KL divergence $\mathrm{KL}(q \,\|\, p)$, which is computationally convenient, but known to suffer from underestimation of the posterior uncertainty. An alternative, which has been our focus here, is the inclusive KL divergence $\mathrm{KL}(p \,\|\, q)$. The benefit of using the inclusive KL is to obtain a more "robust" approximation that does not underestimate the uncertainty. However, in this paper we have argued, and illustrated numerically, that such underestimation of uncertainty can still be an issue, if the optimization is based on *biased* gradient estimates, as is the case for previously proposed VI algorithms. As a remedy, we introduced Markovian score climbing, a new way to reliably learn a variational approximation that minimizes the inclusive KL. This results in a method that melds VI and MCMC. We have illustrated its convergence properties on a simple toy example, and studied its performance on Bayesian probit regression for classification as well as a stochastic volatility model for financial data.

### Broader Impact

MSC is a general purpose approximate statistical inference method. The main goal is to remove systematic errors due to biased estimates of the gradient of the optimization objective function. This can allow for more reliable and robust inferences based on the posterior approximation. However, just like other standard inference methods it does not protect from any bias introduced by applying it to specific models and data [13, 41].

### Acknowledgments and Disclosure of Funding

This work is supported by ONR N00014-17-1-2131, ONR N00014-15-1-2209, NIH 1U01MH115727-01, NSF CCF-1740833, DARPA SD2 FA8750-18-C-0130, Amazon, NVIDIA, and the Simons Foundation. Fredrik Lindsten is financially supported by the Swedish Research Council (project 2016-04278), by the Swedish Foundation for Strategic Research (project ICA16-0015) and by the Wallenberg AI, Autonomous Systems and Software Program (WASP) funded by the Knut and Alice Wallenberg Foundation.

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
