[Supplementary Material]

# A  Supplementary Material

## A.1  Subset Average Likelihood

Li and Turner [35], Dieng et al. [15], who study different classes of divergences where the likelihood also enters as a product, propose to replace the true likelihood at each iteration with a "subset average likelihood". The subset average likelihood approach makes the following approximation

$$p(\mathbf{x}\,|\,\mathbf{z}) = \prod_{i=1}^{n} p(x_i\,|\,\mathbf{z}) \approx p(\mathbf{x}_M\,|\,\mathbf{z})^{\frac{n}{m}} := \prod_{j \in M} p(x_j\,|\,\mathbf{z})^{\frac{n}{m}},$$

where $M \subset \{1, \ldots, n\}$ is a set of indices corresponding to a mini-batch of size $m = |M|$ data points sampled uniformly from $\{x_1, \ldots, x_n\}$ with or without replacement. Considering the same approach for the inclusive KL case the *unbiased* stochastic gradient obtained is

$$\frac{1}{S} \sum_{s=1}^{S} \frac{p(\mathbf{z}^s)p(\mathbf{x}_M\,|\,\mathbf{z}^s)^{\frac{n}{m}}}{q(\mathbf{z}^s\,;\,\lambda)} \nabla_\lambda \log q(\mathbf{z}^s\,;\,\lambda), \ \mathbf{z}^s \sim q(\mathbf{z}\,;\,\lambda). \tag{10}$$

This approximation also leads to a systematic error in the SGD algorithm. It is no longer minimizing the KL divergence from the posterior $p$ to the variational approximation $q$. In fact, it is possible to show that it is actually minimizing the KL divergence from a *perturbed posterior* $\widetilde{p}$, where the likelihood is replaced by a mixture of all potential subset average likelihoods, to the variational approximation $q$. This result is formalized by proposition 2.

**Proposition 2.** *Using the stochastic gradient defined by eq. (10) and an iterative SGD algorithm according to eq. (5) the fixed points $\lambda^\star$ are identical to the solution to*

$$\nabla_\lambda \mathrm{KL}(\widetilde{p}(\mathbf{z}\,|\,\mathbf{x}) \| q(\mathbf{z}\,;\,\lambda)) = 0, \tag{11}$$

*where the perturbed posterior $\widetilde{p}$, if it exists, is given by*

$$\widetilde{p}(\mathbf{z}\,|\,\mathbf{x}) \propto p(\mathbf{z}) \sum_{M \in \mathscr{M}} p(\mathbf{x}_M\,|\,\mathbf{z})^{\frac{n}{m}},$$

*and $\mathscr{M}$ is the set of all possible combinations of mini-batches $M$ of size $m$.*

*Proof.* See the Supplementary Material. □

In the supplement we provide illustrations on a simulated example. It is in general difficult to determine the magnitude of the error introduced by the subset average likelihood in practical applications. The subset average likelihood approach for Rényi and $\chi^2$ divergences [35, 15] likewise leads to a systematic error in the stochastic gradient. Furthermore, the fixed points of the resulting stochastic systems for these divergences are difficult to quantify, making it even harder to understand the effect of the approximation.

**Conditional Sequential Monte Carlo**

Just like CIS is a straightforward modification of IS, so is CSMC a straightforward modification of SMC. We make use of CSMC with ancestor sampling as proposed by Lindsten et al. [38] combined with twisted SMC [23, 25, 47]. While SMC can be adapted to perform inference for almost any probabilistic model [47], we here focus on the state space model

$$p(\mathbf{z}_{1:T}, \mathbf{x}_{1:T}) = p(\mathbf{z}_1)p(\mathbf{x}_1\,|\,\mathbf{z}_1) \prod_{t=1}^{T} p(\mathbf{z}_t\,|\,\mathbf{z}_{t-1})p(\mathbf{x}_t\,|\,\mathbf{z}_t),$$

where we assume that the prior $p(\mathbf{z}_1)$ and transition $p(\mathbf{z}_t\,|\,\mathbf{z}_{t-1})$ are conditionally Gaussian. Because the prior and transition distributions are Gaussian it is convenient to define the full approximation to the posterior $p(\mathbf{z}_{1:T}\,|\,\mathbf{x}_{1:T})$ to be the multivariate normal

$$q(\mathbf{z}_{1:T}\,;\,\lambda) = q(\mathbf{z}_1\,;\,\lambda_1) \prod_{t=2}^{T} q(\mathbf{z}_t\,|\,\mathbf{z}_{t-1}\,;\,\lambda_t), \tag{12}$$

$$q(\mathbf{z}_1\,;\,\lambda_1) \propto p(\mathbf{z}_1)\psi(\mathbf{z}_1\,;\,\lambda_1),$$
$$q(\mathbf{z}_t\,|\,\mathbf{z}_{t-1}\,;\,\lambda_t) \propto p(\mathbf{z}_t\,|\,\mathbf{z}_{t-1})\psi(\mathbf{z}_t\,;\,\lambda_t),$$

where $\psi$ are *twisting potentials*

$$\psi(\mathbf{z}_t \, ; \, \lambda_t) = \exp\left(-\frac{1}{2}\mathbf{z}_t^\top \Lambda_t \mathbf{z}_t + \nu_t^\top \mathbf{z}_t\right),$$

with $\lambda_t = (\Lambda_t, \nu_t)$. We are now equipped to explain the CSMC kernel that updates a conditional trajectory $\mathbf{z}_{1:T}[k-1] = (\mathbf{z}_1[k-1], \ldots \mathbf{z}_T[k-1])$. Each iteration of CSMC consists of three steps: initialization for $t = 1$, running a modified SMC algorithm for $t > 1$, and then updating the conditional sample for the next iteration. We explain in detail below.

First, perform (conditional) IS for the first step where $t = 1$. Set $\mathbf{z}_1^1 = \mathbf{z}_1[k-1]$ and propose the remaining $S-1$ samples from a proposal distribution $q$

$$\mathbf{z}_1^i \sim q(\mathbf{z}_1 \, ; \, \lambda_1), \quad i = 2, \ldots, S$$

and compute the importance weights for $i = 1, \ldots, S$

$$w_1^i = \frac{p(\mathbf{z}_1^i)\psi(\mathbf{z}_1^i \, ; \, \lambda_1)}{q(\mathbf{z}_1^i \, ; \, \lambda_1)}, \qquad\qquad \bar{w}_1^i = \frac{w_1^i}{\sum_{j=1}^{S} w_1^j}.$$

Then, for each step $t > 1$ in turn perform *resampling*, *ancestor sampling*, *propagation* and *weighting*. Resampling picks the most promising earlier sample to propagate, i.e. for $i = 2, \ldots, S$ simulate *ancestor* variables $a_{t-1}^i$ with probability

$$\mathbb{P}\left(a_{t-1}^i = j\right) = \bar{w}_{t-1}^j.$$

For $i = 1$ instead, simulate the corresponding ancestor variable $a_{t-1}^1$ with probability

$$\mathbb{P}\left(a_{t-1}^1 = j\right) \propto \bar{w}_{t-1}^j q(\mathbf{z}_t[k-1] \, | \, \mathbf{z}_{t-1}^j \, ; \, \lambda_t),$$

where $\mathbf{z}_t[k-1]$ is the corresponding element of the conditional trajectory $\mathbf{z}_{1:T}$ from the previous iteration. This is known as ancestor sampling [38].

When propagating for $i = 1$ simply set $\mathbf{z}_t^1 = \mathbf{z}_t[k-1]$, and simulate the remainder from the proposal distribution

$$\mathbf{z}_t^i \sim q(\mathbf{z}_t \, | \, \mathbf{z}_{t-1}^{a_{t-1}^i} \, ; \, \lambda), \quad i = 2, \ldots, S$$

Set $\mathbf{z}_{1:t}^i = (\mathbf{z}_{1:t-1}^{a_{t-1}^i}, \mathbf{z}_t^i)$ and compute the weights for all $i = 1, \ldots, S$

$$w_t^i = \frac{p(\mathbf{z}_t^i \, | \, \mathbf{z}_{t-1}^{a_{t-1}^i})p(\mathbf{x}_{t-1} \, | \, \mathbf{z}_{t-1}^{a_{t-1}^i})}{q(\mathbf{z}_t^i \, | \, \mathbf{z}_{t-1}^{a_{t-1}^i} \, ; \, \lambda_t)} \frac{\psi(\mathbf{z}_t^i \, ; \, \lambda_t)}{\psi(\mathbf{z}_{t-1}^{a_{t-1}^i} \, ; \, \lambda_{t-1})}, \tag{13}$$

$$\bar{w}_t^i = \frac{w_t^i}{\sum_{j=1}^{S} w_t^j}.$$

For the final step $t = T$ the (unnormalized) weights are instead

$$w_T^i = \frac{p(\mathbf{z}_T^i \, | \, \mathbf{z}_{T-1}^{a_{T-1}^i})p(\mathbf{x}_{T-1} \, | \, \mathbf{z}_{T-1}^{a_{T-1}^i})}{q(\mathbf{z}_T^i \, | \, \mathbf{z}_{T-1}^{a_{T-1}^i} \, ; \, \lambda_T)} \frac{p(\mathbf{x}_T \, | \, \mathbf{z}_T^i)}{\psi(\mathbf{z}_{t-1}^{a_{t-1}^i} \, ; \, \lambda_{t-1})}. \tag{14}$$

Finally, an updated conditional sample is generated by picking one of the proposed trajectories with probability proportional to its (normalized) weight, i.e.

$$\mathbf{z}_{1:T}[k] = \mathbf{z}_{1:T}^J,$$

where $J$ is a discrete random variable with probability $\mathbb{P}(J = j) = \bar{w}_T^j$.

Repeating this procedure iteratively constructs a Markov chain with the posterior $p(\mathbf{z}_{1:T} \, | \, \mathbf{x}_{1:T})$ as its stationary distribution [2, 38, 47]. With this it is possible to attain an estimate of the gradient with respect to the variational parameters of eq. (2) as follows

$$\widehat{g}_{\mathrm{KL}}(\lambda) = -\nabla_\lambda \log q(\mathbf{z}_{1:T}[k] \, ; \, \lambda), \tag{15}$$

where $\mathbf{z}_{1:T}[k]$ is the conditional sample retained at iteration $k$ of the CSMC algorithm.

We summarize one full iteration of the CSMC algorithm in algorithm 4. This algorithm defines a Markov kernel $M(\mathbf{z}_{1:T}[k] \, | \, \mathbf{z}_{1:T}[k-1] \, ; \, \lambda)$ useful for MSC.

**Algorithm 4:** Conditional Sequential Monte Carlo

**Input** : Model $p(\mathbf{z}_{1:T}, \mathbf{x}_{1:T})$, proposal $q(\mathbf{z}_{1:T} \,;\, \lambda)$, conditional sample $\mathbf{z}_{1:T}[k-1]$, and total number of internal samples $S$.

**Output** : $\mathbf{z}_{1:T}[k] \sim M(\cdot \,|\, \mathbf{z}_{1:T}[k-1] \,;\, \lambda)$, updated conditional sample.

1 Set $\mathbf{z}_1^1 = \mathbf{z}_1[k]$
2 Sample $\mathbf{z}_1^i \sim q(\mathbf{z}_1 \,;\, \lambda_1)$ for $i = 2, \ldots, S$
3 Compute $w_1^i = \dfrac{p(\mathbf{z}_1^i)\psi(\mathbf{z}_1^i \,;\, \lambda_1)}{q(\mathbf{z}_1^i \,;\, \lambda_1)}$ for $i = 1, \ldots, S$
4 **for** $t = 2, \ldots, T$ **do**
5    **for** $i = 2, \ldots, S$ **do**
6       Sample $a_{t-1}^i$ w.p. $\mathbb{P}(a_{t-1}^i = j) = \bar{w}_{t-1}^j$
7       Sample $\mathbf{z}_t^i \sim q(\mathbf{z}_t \,|\, \mathbf{z}_{t-1}^{a_{t-1}^i} \,;\, \lambda_t)$
8    **end**
9    Sample $a_{t-1}^1$ w.p.
10    $\mathbb{P}(a_{t-1}^i = j) \propto \bar{w}_{t-1}^j q(\mathbf{z}_t[k] \,|\, \mathbf{z}_{t-1}^j \,;\, \lambda_t)$
11    Set $\mathbf{z}_t^1 = \mathbf{z}_t[k]$
12    **for** $i = 1, \ldots, S$ **do**
13       Compute $w_t^i$ in eq. (13) or eq. (14)
14       Set $\mathbf{z}_{1:t}^i = \left( \mathbf{z}_{1:t-1}^{a_{t-1}^i}, \mathbf{z}_t^i \right)$
15    **end**
16 **end**
17 Sample $J$ with probability $\mathbb{P}(J = j) \propto \bar{w}_T^j$
18 Set $\mathbf{z}_{1:T}[k] = \mathbf{z}_{1:T}^J$

**Proof of Proposition 1**

This result is an adaptation of Gu and Kong [21, Theorem 1] based on Benveniste et al. [5, Theorem 3.17, page 304]. Let $\lambda^\star$ be a minimizer of the inclusive KL divergence in eq. (2). Consider the ordinary differential equation (ODE) defined by

$$\frac{\mathrm{d}}{\mathrm{d}t}\lambda(t) = \mathbb{E}_{p(\mathbf{z}|\mathbf{x})}[-s(\mathbf{z} \,;\, \lambda(t))], \quad \lambda(0) = \lambda_0, \tag{16}$$

and its solution $\lambda(t)$, $t \geq 0$. If the ODE in eq. (16) admits the unique solution $\lambda(t) = \widehat{\lambda}$, $t \geq 0$ for $\lambda(0) = \widehat{\lambda}$, then $\widehat{\lambda}$ is called a stability point. The minimizer $\lambda^\star$ is a stability point of eq. (16). A set $\Lambda$ is called the domain of attraction of $\widehat{\lambda}$, if the solution to eq. (16) for $\lambda(0) \in \Lambda$ remains in $\Lambda$ and converges to $\widehat{\lambda}$. Suppose that $\lambda_k \in \mathbb{R}^d$ and that $\Lambda$ is an open set in $\mathbb{R}^{d_\lambda}$. Furthermore, suppose $\mathbf{z}[k] \in \mathbb{R}^{d_z}$ and that $Z$ is an open set in $\mathbb{R}^{d_z}$. Denote the Markov kernel in MSC, algorithm 1, by $M_\lambda(\mathbf{z}, \mathrm{d}\mathbf{z}')$ and repeated application of it by $M_\lambda^k(\mathbf{z}, \mathrm{d}\mathbf{z}') = \int \cdots \int M_\lambda(\mathbf{z}, \mathrm{d}\mathbf{z}_1) M_\lambda(\mathbf{z}_2, \mathrm{d}\mathbf{z}_3) \cdots M_\lambda(\mathbf{z}_{k-1}, \mathrm{d}\mathbf{z}')$. $|\mathbf{z}|$ denotes the length of the vector $\mathbf{z}$. Let $Q$ be any compact subset of $\Lambda$, and $q > 1$ a sufficiently large real number such that the following assumptions hold. We follow Gu and Kong [21] and assume:

**C 1.** *Assume that the step size sequence satisfies* $\sum_{k=1}^\infty \varepsilon_k = \infty$ *and* $\sum_{k=1}^\infty \varepsilon_k^2 < \infty$.

**C 2** (Integrability). *There exists a constant $C_1$ such that for any $\lambda \in \Lambda$, $\mathbf{z} \in Z$ and $k \geq 1$,*

$$\int \left(1 + |\mathbf{z}'|^q\right) M_\lambda^k(\mathbf{z}, \mathrm{d}\mathbf{z}') \leq C_1 \left(1 + |\mathbf{z}|^q\right)$$

**C 3** (Convergence of the Markov Chain). *Let $p(\mathbf{z}|\mathbf{x})$ be the unique invariant measure for $M_\lambda$. For each $\lambda \in \Lambda$,*

$$\lim_{k \to \infty} \sup_{\mathbf{z} \in Z} \frac{1}{1 + |\mathbf{z}|^q} \int \left(1 + |\mathbf{z}'|^q\right) |M_\lambda^k(\mathbf{z}, \mathrm{d}\mathbf{z}') - p(\mathrm{d}\mathbf{z}'|\mathbf{x})| = 0.$$

**C 4** (Continuity in $\lambda$). *There exists a constant $C_2$, such that for all $\lambda, \lambda' \in Q$*

$$\left| \int \left(1 + |\mathbf{z}'|^q\right)\left(M_\lambda(\mathbf{z}, d\mathbf{z}') - M_{\lambda'}(\mathbf{z}, d\mathbf{z}')\right) \right| \leq C_2 |\lambda - \lambda'|\left(1 + |\mathbf{z}|^q\right).$$

**C 5** (Continuity in $\mathbf{z}$). *There exists a constant $C_3$, such that for all $\mathbf{z}_1, \mathbf{z}_2 \in Z$*

$$\sup_{\lambda \in \Lambda} \left| \int \left(1 + |\mathbf{z}'|^{q+1}\right)\left(M_\lambda(\mathbf{z}_1, d\mathbf{z}') - M_\lambda(\mathbf{z}_2, d\mathbf{z}')\right) \right| \leq C_3 |\mathbf{z}_1 - \mathbf{z}_2|\left(1 + |\mathbf{z}_1|^q + |\mathbf{z}_2|^q\right).$$

**C 6** (Conditions on the Score Function). *For any compact subset $Q \subset \Lambda$, there exist positive constants $p, K_1, K_2, K_3$ and $\nu > 1/2$ such that for all $\lambda, \lambda' \in \Lambda$ and $\mathbf{z}, \mathbf{z}_1, \mathbf{z}_2 \in Z$,*

$$|\nabla_\lambda \log q(\mathbf{z}; \lambda)| \leq K_1\left(1 + |\mathbf{z}|^{p+1}\right),$$
$$|\nabla_\lambda \log q(\mathbf{z}_1; \lambda) - \nabla_\lambda \log q(\mathbf{z}_2; \lambda)| \leq K_2 |\mathbf{z}_1 - \mathbf{z}_2|(1 + |\mathbf{z}_1|^p + |\mathbf{z}_2|^p),$$
$$|\nabla_\lambda \log q(\mathbf{z}; \lambda) - \nabla_\lambda \log q(\mathbf{z}; \lambda')| \leq K_3 |\lambda - \lambda'|^\nu \left(1 + |\mathbf{z}|^{p+1}\right).$$

The constants $C_1, \ldots, C_3$ and $\nu$ may depend on the compact set $Q$ and the real number $q$. C 1 is the standard Robbins-Monro condition and C 6 controls the regularity of the model. C 2-C 5 have to do with the convergence and continuity of the Markov kernel. These conditions can be difficult to verify in the general case, but can be proven more easily under the simplifying assumption that Z is compact. See Lindholm and Lindsten [37, Appendix B] for a proof of continuity of the CSMC kernel, which can also be adapted to the CIS kernel.

With the above assumptions the result follows from Gu and Kong [21, Theorem 1] where (left - their notation, right - our notation)

$$\theta = \lambda,$$
$$x = \mathbf{z},$$
$$\Pi_\theta = M_\lambda,$$
$$H(\theta, x) = \nabla_\lambda \log q(\mathbf{z}; \lambda),$$

and $I(\theta, x) = 0$, $\Gamma_k = 0$.

**Proof of Proposition 2**

The fixed points of the iterative algorithm are the solutions to the equation when we set the expectation of eq. (10) equal to zero. The equation is given by

$$\mathbb{E}\left[-\frac{1}{S}\sum_{s=1}^{S}\frac{p(\mathbf{z}^s)p(\mathbf{x}_M \mid \mathbf{z}^s)^{\frac{n}{m}}}{q(\mathbf{z}^s; \lambda)}\nabla_\lambda \log q(\mathbf{z}^s; \lambda)\right] = \mathbb{E}\left[-\frac{p(\mathbf{z})p(\mathbf{x}_M \mid \mathbf{z})^{\frac{n}{m}}}{q(\mathbf{z}; \lambda)}\nabla_\lambda \log q(\mathbf{z}; \lambda)\right]$$

$$= \mathbb{E}\left[-\frac{p(\mathbf{z})\frac{1}{|\mathcal{M}|}\sum_{M \in \mathcal{M}} p(\mathbf{x}_M \mid \mathbf{z})^{\frac{n}{m}}}{q(\mathbf{z}; \lambda)}\nabla_\lambda \log q(\mathbf{z}; \lambda)\right] = 0$$

$$\iff \mathbb{E}_{\widetilde{p}(\mathbf{z}|\mathbf{x})}[-\nabla_\lambda \log q(\mathbf{z}; \lambda)] = 0$$

$$\iff \nabla_\lambda \mathrm{KL}(\widetilde{p}(\mathbf{z} \mid \mathbf{x})\|q(\mathbf{z}; \lambda)) = 0,$$

where the first equality follows because the samples $\mathbf{z}^s$ are independent and identically distributed. The second equality follows by the distribution of the mini-batches. The first equivalence follows because $\mathbf{z} \sim q(\mathbf{z}; \lambda)$ and we multiply both sides by a constant independent of $\lambda$. The final equivalence follows because $\widetilde{p}(\mathbf{z} \mid \mathbf{x})$ does not depend on $\lambda$. This concludes the proof.

**Additional Results Bayesian Probit Regression**

We also compare the posterior uncertainty learnt using MSC and IS. Figure 4 shows difference in the log-standard deviation between the posterior approximation learnt using MSC and that using IS, i.e. $\log \sigma^\star_{\mathrm{MSC}} - \log \sigma^\star_{\mathrm{IS}}$. The figure contains one boxplot for each dimension of the latent variable and is based on data from 100 random train-test splits. We can see that for two of the datasets, Heart and Ionos, MSC on average learns a posterior approximation with higher uncertainty. However, for the Pima dataset the IS-based method tends to learn higher variance approximations.

**(a)** Pima          **(b)** Ionos          **(c)** Heart

**Figure 4:** The difference in log-standard deviation of the variational approximation, $\log \sigma^{\star}_{\text{MSC}} - \log \sigma^{\star}_{\text{IS}}$, between parameters learnt using MSC (this paper) and IS (cf. [7]). The dimension of the latent variable is plotted versus the parameters learnt from 100 random splits.

**Additional Results Subset Average Likelihoods**

We illustrate the difference between the true and perturbed posteriors in fig. 5 for a toy example where the two distributions can be computed exactly. The model is an unknown mean measured in Gaussian noise with a conjugate prior, i.e. $\mathbf{z} \sim \mathcal{N}(0, 1)$, $x_i \sim \mathcal{N}(\mathbf{z}, 1)$. To be able to exactly compute the perturbed posterior we keep the number of data points small $n = 10$. The figure shows the true and perturbed posteriors for two randomly generated datasets with $m = 2, 5, 9$.

**Bias in $\chi$-divergence variational inference (CHIVI)**

Figure 6 illustrates the systematic error introduced in the optimal parameters of CHIVI when using biased gradients.

**Figure 5:** Example of perturbed and true posterior when using subset average likelihoods. The data used is simulated from the model defined by $\mathbf{z} \sim \mathcal{N}(0, 1)$, $x_i \sim \mathcal{N}(\mathbf{z}, 1)$, $n = 10$ for two different random seeds. The subset sizes where chosen to be $m = 2$ (top row), $m = 5$ (top row) and $m = 9$ (bottom row).

**Figure 6:** Example of learnt variational parameters for CHIVI, as well as true parameters when using a Gaussian approximation to a skew normal posterior distribution.