[Reviews · NeurIPS 2020]

Review 1

Summary and Contributions: This paper proposes a method to perform Variational Inference (VI) using the inclusive Kullback-Leibler divergence KL(p||q), with q denoting the variational approximation. Compared to the predominant exclusive KL(q||p), the inclusive KL does not suffer from the underestimation of the posterior uncertainty. The latter is however less convenient to work with, as it involves an expectation of the score function w.r.t. the true (intractable) posterior. The authors argue that existing solutions are either biased or suffer from high variance, which may result in estimated parameters that are no longer optimal with respect to the corresponding minimization problem. To tackle this issue and deal with the intractable expectation, the proposed method uses stochastic gradients estimated using samples from an MCMC algorithm, whose stationary distribution is the true posterior and proposal distribution is the variational approximation. Compared to some alternatives, the proposed algorithm exhibits better convergence properties as illustrated both empirically and theoretically.

Strengths: This is a well-written and clearly presented paper. Relying on the inclusive KL for inference is not new by itself, but the proposed method to minimize it, although simple, has not been investigated in this context before. The resulting algorithm is technically sound and exhibits better convergence properties compared to existing alternatives, as supported empirically and theoretically in the asymptotic regime.

Weaknesses: As discussed in the paper one weakness of the proposed method is scalability. While the authors discuss some possible solutions, this may limit the adoption of the proposed algorithm in practice, since one key advantage for relying on VI methods is the possibility to scale them to large datasets when combined with stochastic optimization. While the inclusive KL(p||q) does not suffer from the underestimation of the posterior variance, it may result in variational approximations q stretching to cover all p to the expense of putting non-negligible probability mass in regions where the posterior is close or even equal to zero. This may lead to poor predictive distributions in some situations, such as when q is under specified. The paper can benefit from including a discussion about this undesirable behavior, as well as characterizing situations in which KL(p||q) may be more suitable than KL(q||p) and vice versa.

Correctness: The claims, method and empirical methodology are correct and sound to the extent I checked.

Clarity: The paper is well written and easy to follow.

Relation to Prior Work: The contribution of this paper and its relation to existing work are clearly stated.

Reproducibility: Yes

Additional Feedback: Please refer to the weaknesses section. *** Post Rebuttal Update *** I thank the authors for their response, which addresses my concerns. I also went through the other reviews, and my original assessment remains the same.


Review 2

Summary and Contributions: The paper proposes a novel variation of variational inference, based upon KL(posterior|approx) instead of the usual mode-seeking KL(approx|posteror). The method is based on a combination of MCMC/SMC techniques and stochastic gradients. The main idea, following [19], is to use a conditional Markov transition kernel to obtain an iteratively re-fined stochastic approximation for the necessary gradients in the KL loss. The method is shown to converge to a local minimum of the loss. An empirical evaluation shows benefits on a number of smaller benchmark tasks.

Strengths: The proposed methodology is very appealing in terms of its properties: the inclusive KL term has been observed to yield in some cases much better uncertainty estimates that the mode-seeking KL. Yet, it resisted many efforts of constructing algorithms that are tractable and converge to something meaningful, going back all the way to convergence issues in the early works on EP. Using conditional MC kernels as a way to avoid running expensive (and biased) MCMC chains from scratch to estimate gradients independently in every iteration is a very worthwhile contribution, and the construction is indeed elegant. The fact that this framework allows for provable convergence is remarkable (though not original work, but an application of [19]) The (limited) empirical evaluation shows clear improvements against both EP and recent modern variational inference variations: less bias, more stability, better uncertainty estimation.

Weaknesses: There are a number of points which in my opinion significantly weaken the paper: the unsuitability for large datasets (while maintaining guarantees), and the lack of more interesting empirical applications (2/3 of the experiments are essentially straw-men) The first issue is noted by the authors, and further studied in the supplementary material: Subsampling approximations for KL(p,q) are not at all straight-forward, re-introducing the systematic errors that the method sought to overcome. I have strong doubts that any introduced approximation will results in a stable, generally useful algorithm. The authors note that techniques from MCMC for tall datasets (with guarantees) might be the rescue, but I would argue that these methods either don't work on anything that extremely-well-specified problems, or require almost all of the available data (this is essentially the take-home message of [4]). The second issue in my opinion is easily rectified with a more sensible choice of applications. The skew-normal example is designed to break the other (biased gradient) methods, and does so successfully, but this is essentially a straw man. The probit regression example is the other extreme -- a posterior that looks almost like a Gaussian distribution, and pretty much any inference method will work reasonably well.

Correctness: As the method is an instantiation of [19], the proof is an instantiation of the necessary conditions.

Clarity: The paper is very well presented and written overall. A few points that would improve clarity would be: * a few comments on the nature of the conditions 1-6 in the appendix * an intuitive explanation (or even proof overview) of why the variational parameters converge. In particular, how are oscillations between z and \lambda avoided? Is the decreasing step-size responsible for this?

Relation to Prior Work: The paper applies a previous algorithm within the context of VI and clearly states that. There is a nice discussion of related relevant methodology, with comments on strengths and weaknesses.

Reproducibility: Yes

Additional Feedback: ################################ Thanks to the authors for the thoughtful rebuttal. I think an example of the approach for running this algorithm on larger datasets would benefit the paper (as the other reviewers also pointed this out).


Review 3

Summary and Contributions: The authors show that a combination of Monte Carlo and variational inference (minimizing the inclusive KL(p||q) rather than the more traditional exclusive KL(q||p)) allows us to obtain a) unbiased samples from the posterior over the hidden variables; b) unbiased samples from the gradient of the divergence KL(p||q) wrt the variational parameters, and c) unbiased samples from the gradient of the likelihood wrt generative parameters. Another good way to summarize this contribution is to say that it is identical to the reweighted wake-sleep (RWS) algorithm, but with a slight modification on the sampling that makes the estimations unbiased.

Strengths: The paper pursues a goal with far reaching applications: approximate inference and learning in intractable Bayesian models. Achieving both low-variance and unbiased gradients (which in turn enables ML learning) can enable the use of more sophisticated models than was previously possible. The resulting algorithm is also pretty simple to implement and is pretty much standard, almost identical to RWS. The authors explicitly state the challenges involved in applying this algorithm to large training sets and detail why subset average likelihoods do not solve this problem. Being clear about this limitation from the onset is very valuable. The paper is written in a very clear and educational format and is easy to read.

Weaknesses: The stated inability to deal with large datasets (i.e., no obvious way to make use of minibatches while keeping exactness) in a very important practical limitation. While unbiased gradients are nice to have, biased gradients, such as those obtained by RWS, are in practice good enough when the number of samples is sufficient for the bias not to be too large. The authors argue that that requires selecting an appropriate number of samples S, but that's usually a parameter that is explored anyway to make sure that the variance of the gradient is small enough, so it's unlikely to be a problem in practice. The experimental section shows very limited practical benefit from the use of this proposal wrt RWS. The last experiment does not test against RWS, why? Standard variational Bayes has a single maximization target to be optimized wrt both generative and variational parameters, whereas the presented formulation performs separate gradient approximations with no single cost functional. This is not a limitation of the current proposal, but does limit further developments on top of it (e.g., by leveraging more sophisticated joint optimizers). I think that these weaknesses are significant from a practical perspective, but I believe the theoretical contribution and the proof of unbiased convergence are useful enough to justify this paper.

Correctness: The results presented in the main paper are correct.

Clarity: The presentation keeps mixing RWS and IS as if they were they same thing. IS is even claimed to be a biased estimator, when in fact it is not. RWS (eq. 7) is biased, IS (eq. 6) is not. I'd suggest to change the use of the term "IS" throughput the paper to either RWS or "normalized IS", so that it is clear that you are not, in fact, talking about IS. Further explanations about CIS and CSMC would be helpful. Are the samples z exact uncorrelated samples from the posterior? Since CIS and CSMC are in my understanding the main contributions of this work (the other aspects are pretty much standard), I would like to see more commentary on the reasoning behind the proposed modification and some intuition on why it leads to an unbiased estimator. Also, it is my understanding that the samples of the gradient (both over lambda and theta) in Algorithm 3 are unbiased. However, the claim in Proposition 1 seems weaker than that. Can the authors clarify if the gradients are unbiased?

Relation to Prior Work: Prior work is properly discussed.

Reproducibility: Yes

Additional Feedback: The submitted PDF is full of hyperlinks in blue for multiple terms. Those links just take us to the first page of the paper, so they seem to offer no value and are just distracting.


Review 4

Summary and Contributions: The authors develop Markovian Score Climbing (MSC) an algorithm that minimized the inclusive KL-divergence. To this end the authors propose to repeatedly sample from Markov kernels whose stationary distribution is the posterior. The corresponding Markov kernels are constructed using twisted CSMC. The authors theoretically show that MSC's estimates of the variational parameters are consistent and empirically verify that its convergence properties on a simple toy example. They further evaluate their methods on a Bayesian probit regression task and stochastic volatility model and find that MSC performs competitively compared to their baselines, IS/SMC and EP.

Strengths: The proposed algorithm provides provable consistent estimates of the parameters of the variational distribution

Weaknesses: It is not surprising that IS struggles and is highly biased with low number of samples. I assume that increasing the number of samples but decreasing the number of times steps would negatively impact the performance of MSC (similar to other MCMC methods), as the MC may not converge fast enough.The experiments do not evaluate such a trade-off at all? If I have not missed something the number of samples in the first experiment is 2 and the algorithm was run for 10^6 iterations. In the second experiment the number of samples varies from 5 to 50 with an unknown number of time steps, and in the third experiment the number of samples is 10 again the number of time steps is missing. In my opinion there needs to be an evaluation with a fixed sampling budget, i.e. a fixed number of sampling operations (num_paticles*mc_lenght). Further the empirical evaluation only considers EP and naive importance sampling. It would be great to see the performance of some stronger baselines. For example, RWS can be augmented with Hamiltonian Monte Carlo updates analogous to the method proposed in [1]. [1] Hoffman, M. D. Learning deep latent gaussian models with markov chain monte carlo. In Proceedings of the 34th International Conference on Machine Learning-Volume 70, pp. 1510–1519. JMLR. org, 2017.

Correctness: The claim and empirical methodology are to the best of my knowledge correct.

Clarity: I found the writing to be slightly repetitive and somewhat confusing at times. E.g. on line 38 - 41 the authors write: "However, these methods involve biased gradients of the inclusive KL, which leads to bias in their final estimates. In contrast, MSC provides consistent gradients for essentially no added cost while providing better variational approximations" The statement is quite confusing as it compares different properties of a gradient estimator without further qualification. While, say self-normalized IS is biased it is still consistent (w.r.t. the number of samples), i.e. in the infinite limit of samples the self-normalized IS estimate converges a.s. to the true expected gradient. MSC's gradient estimates are consistent as well but in a different sense (w.r.t. the number of time steps), i.e. as the MC converges to its stationary distribution the gradient estimate converges to the true expected gradient and so does the estimate of the variational parameters. (Please correct me if I misinterpreting something here) In my opinion this is an important point and should be discussed more clearly. The introduction and background section are very lengthy while interesting information about the construction of the MC-kernel via twisted CSMC is not part of the main paper but exists in the supplementary material.

Relation to Prior Work: Prior work is discussed but some important related work is missing. I listed some related work which in my opinion should be discussed below. Wang and colleges [1] develop a meta-learning approach to learn Gibbs block conditionals. The paper has a different focus and assumes to have access to samples from the true generative model but is still technically related. They optimize an inclusive KL-divergence and employ additional MH steps to maintain the correct stationary distribution. Li et al. [2] proposes to use MCMC to improve the quality of samples from an initial encoder while using them to minimize an inclusive KL divergence relative to the filtering distributions of the markov chain. While they are mainly interested in learning a good proposals and do not consider learning good MCMC kernels their general framework seems to be very close to the one described in the paper. The authors should definitely contrast their work with the work presented in this paper. Most recently published (but it has been on arXiv for a while) Wu and colleges [3] proposed to minimize the inclusive KL-divergence to learn block proposal which approximate conditional gibbs updates. These proposals can be applied repetitively in a SMC-sampler framework to improve the quality of the initial sample. [1] Wang, T., Wu, Y., Moore, D., and Russell, S. J. Metalearning mcmc proposals. In Advances in Neural Information Processing Systems, pp. 4146–4156, 2018 [2] Li, Y., Turner, R. E., and Liu, Q. Approximate inference with amortised mcmc. arXiv preprint arXiv:1702.08343, 2017. [3] Wu, H., Zimmermann, H., Sennesh, E., Le T.A., and van de Meent, J. Amortized Population Gibbs Samplers with Neural Sufficient Statistics. International Conference on Machine Learning (ICML), 2020.

Reproducibility: Yes

Additional Feedback: **Update** I've carefully read through the author's rebuttal and the other reviews. The authors addressed all of my concerns in their rebuttal and I'm happy about their decision to add a more clear discussion regarding the properties of their method compared to standard RWS/IS and to add the missing related work. However, the trade-off between sample size S and time steps/iterations K is still not clear to me. Specifically, it is not clear to me how a reduction of the number of steps while increasing the numbers of samples per step under a fixed overall sampling budget affects the performance of MSC compared to self-normalized IS variants. I understand that it is straightforward to see how the performance of these algorithms changes when considering only one of these dimensions (steps or samples), but without any experimentation or analysis I can not see how their performance changes relative to each other when considering both dimensions. In any case an evaluation of their method under different allocations of a fixed sampling budgets would be something I'd be interesting in to see. Also, while I do see why improving their baselines by employing additional HMC steps does not contribute to highlighting the differences between their framework and RWS/IS. A demonstration of the performance of the HMC-augmented baseline would still be interesting and highlight the strength of their methods if MSC performs similarly or even better without augmentation. Given the rebuttal and discussion I increase my score to a 6

[Author Response · NeurIPS 2020]

We thank the reviewers for the constructive feedback, which will significantly improve the paper. Reviewers 1-3 all
agree that it is a well-written paper and a worthy contribution to the conference. One common weakness discussed was
data scalability. We elaborate on this first and address specific comments and questions from the reviewers below.

**On the scalability of MSC**  Variational inference based on KL(q‖p) is scalable in the sense that it works by subsam-
pling datasets both for *exchangeable data*, $p(x_{1:n}) = \mathbb{E}_{p(z)}\left[\prod_{i=1}^{n} p(x_i|z)\right]$, as well as for *independent and identically*
*distributed data* (iid), $p(x_{1:n}) = \prod_{i=1}^{n} p(x_i) = \prod_{i=1}^{n} \mathbb{E}_{p(z_i)}[p(x_i|z_i)]$. Often in the literature (such as for VAEs,
RWS, etc.) applications assumes the data is generated iid and and achieve scalability through use of subsampling and
amortization. The current discussion in Section 3.5 for MSC on the other hand focuses on the more challenging case,
when the data is exchangeable. In fact, MSC can potentially scale just as well as other algorithms to large datasets
when data is assumed iid $x_i \sim p(x)$, $i = 1, \ldots, n$. Instead of minimizing $\mathrm{KL}(p(z|x)\|q(z;\lambda))$ wrt $\lambda$ for each $x = x_i$,
we consider minimizing $\mathrm{KL}(p(x)p(z|x)\|p(x)q(z|\lambda_\eta(x)))$ wrt $\eta$ where $\lambda_\eta(x)$ is an inference network (amortization).
If $q(z|\lambda_\eta(x))$ is flexible enough the posterior $p(z|x)$ is the optimal solution to this minimization problem. Stochastic
gradient descent can be performed by noting that

$$\nabla_\eta \mathrm{KL}(p(x)p(z|x)\|p(x)q(z|\lambda_\eta(x))) = 0 + \mathbb{E}_{p(x)p(z|x)}\left[-\nabla_\eta \log q(z|\lambda_\eta(x)\right] \approx \frac{1}{n}\sum_{i=1}^{n} \mathbb{E}_{p(z|x_i)}\left[-\nabla_\eta \log q(z|\lambda_\eta(x_i))\right],$$

where the right hand side is directly amenable to data subsampling. We will include a discussion of this aspect of
scalability in Section 3.5 in the revision.

**Reviewer 4: Consistency of self-normalized IS vs MSC**  MSC *does not* require the MCMC sampler to reach
stationarity at each iteration k of Alg 1. A single update (line 2) with any initialization ensures convergence as the
number of iterations k increases. In contrast self-normalized IS would require an infinite number of samples *at each*
*iteration k* to ensure convergence to a minima of KL(p‖q). That is, self-normalized IS/RWS is "doubly asymptotic"—
both the number of samples per iteration and the number of iterations need to go to infinity for convergence. MSC, on
the other hand, is "singly asymptotic"—the MCMC iterations are intertwined with the optimization iterations k and will
converge as k goes to infinity even with a single MCMC step per iteration. We will clarify this in the revision.

**Reviewer 4: Trade-off between sample size S and time steps/iterations K**  The iterations, or time steps, corre-
sponds to regular parameter updates in any gradient descent algorithm. The second experiment used 10,000 iterations
for both RWS and MSC as very little change in parameter estimates was observed for longer runs. Increasing the
number of samples S would not negatively impact the performance of MSC compared to self-normalized IS variants
(see eg Fig 2), as the Rao-Blackwellization (below eq 8) would similarly improve the gradient estimate quality for MSC.

**Reviewer 4: Stronger baselines, RWS with HMC**  We compare the base versions of the respective algorithms. Just
like RWS can be improved by introducing HMC updates, so can a similar benefit be achieved by introducing such
updates in the MSC Markov kernel.

**Reviewer 4: Missing related work**  We will add these references to the related work section. The most similar is
Li et al. (2017). The main difference compared to MSC is that Li et al. seek to minimize $\mathrm{D}\left(\mathcal{K}_T q(z;\lambda)\|q(z;\lambda)\right)$,
where $\mathcal{K}_T$ corresponds to an MCMC kernel $\mathcal{K}$ with stationary distribution $p(z|x)$ applied $T$ times. Because they
re-initialize the MCMC procedure at each update, corresponding to $\mathcal{K}_T q(z;\lambda)$, this will not converge to a minimum of
$\mathrm{D}\left(p(z|x)\|q(z;\lambda)\right)$. This in contrast to MSC that provably minimizes $\mathrm{D}\left(p(z|x)\|q(z;\lambda)\right)$ directly.

**Reviewer 3: The last experiment does not test against RWS, why?**  RWS is based on self-normalized IS that
performs poorly for state space models. We compare instead to neural adaptive SMC, an algorithm based on SMC
tailored to state space models, which is a stronger baseline than RWS for this application.

**Reviewer 3: CIS/CSMC, unbiasedness and convergence**  The gradients estimated by CIS/CSMC are not unbiased,
but due to their Markov properties the bias vanishes as the number of iterations K increases. This means it is still
possible to show convergence to a true optima of the inclusive KL as shown by Prop 1 and as illustrated empirically
in Fig 1. This is in contrast to self-normalized IS/SMC-based gradients whose bias does not vanish as the number of
iterations K increases, also illustrated in Fig 1. Contrary to MSC, the self-normalized IS/SMC-based algorithms require
an infinite number of samples at *each iteration* to ensure convergence, as discussed above.

**Reviewer 3: Mixing up self-normalized IS/RWS and IS**  We will revise the notation to make this difference clear.

**Reviewer 2: Conditions 1-6 and convergence intuition**  The assumptions are fairly standard for Markovian stochas-
tic approximations. The conditions on the MCMC kernel can be verified for CIS/CSMC under compactness assumptions.
We will add discussion and references to the supplement.

**Reviewer 1: Predictive distributions based on KL(p‖q)**  We will revise and include this aspect of KL(p‖q) as well
as an extended discussion on when one is preferable to the other.

[Meta-Review · NeurIPS 2020]

This work proposes a novel variation for VI, based on a combination of MCMC/SMC and stochastic gradients. The key idea is using a conditional Markov transition kernel to obtain increasingly refined estimates of the KL gradients. The empirical results are provided on smaller datasets and it has been pointed out that the paper would improve, if scalability of the method could have been illustrated via experiments on larger datasets.